# Mixing Time Estimation in Reversible Markov Chains from a Single Sample Path

**Daniel Hsu**
Columbia University
djhsu@cs.columbia.edu

**Aryeh Kontorovich**
Ben-Gurion University
karyeh@cs.bgu.ac.il

**Csaba Szepesvári**
University of Alberta
szepesva@cs.ualberta.ca

## Abstract

This article provides the first procedure for computing a fully data-dependent interval that traps the mixing time $t_{\mathrm{mix}}$ of a finite reversible ergodic Markov chain at a prescribed confidence level. The interval is computed from a single finite-length sample path from the Markov chain, and does not require the knowledge of any parameters of the chain. This stands in contrast to previous approaches, which either only provide point estimates, or require a reset mechanism, or additional prior knowledge. The interval is constructed around the relaxation time $t_{\mathrm{relax}}$, which is strongly related to the mixing time, and the width of the interval converges to zero roughly at a $\sqrt{n}$ rate, where $n$ is the length of the sample path. Upper and lower bounds are given on the number of samples required to achieve constant-factor multiplicative accuracy. The lower bounds indicate that, unless further restrictions are placed on the chain, no procedure can achieve this accuracy level before seeing each state at least $\Omega(t_{\mathrm{relax}})$ times on the average. Finally, future directions of research are identified.

## 1 Introduction

This work tackles the challenge of constructing fully empirical bounds on the mixing time of Markov chains based on a single sample path. Let $(X_t)_{t=1,2,\dots}$ be an irreducible, aperiodic time-homogeneous Markov chain on a finite state space $[d] := \{1, 2, \dots, d\}$ with transition matrix $\boldsymbol{P}$. Under this assumption, the chain converges to its unique stationary distribution $\boldsymbol{\pi} = (\pi_i)_{i=1}^d$ regardless of the initial state distribution $\boldsymbol{q}$:

$$\lim_{t \to \infty} \Pr_{\boldsymbol{q}} (X_t = i) = \lim_{t \to \infty} (\boldsymbol{q}\boldsymbol{P}^t)_i = \pi_i \quad \text{for each } i \in [d].$$

The *mixing time* $t_{\mathrm{mix}}$ of the Markov chain is the number of time steps required for the chain to be within a fixed threshold of its stationary distribution:

$$t_{\mathrm{mix}} := \min \left\{ t \in \mathbb{N} : \sup_{\boldsymbol{q}} \max_{A \subset [d]} |\Pr_{\boldsymbol{q}} (X_t \in A) - \boldsymbol{\pi}(A)| \leq 1/4 \right\}. \quad (1)$$

Here, $\boldsymbol{\pi}(A) = \sum_{i \in A} \pi_i$ is the probability assigned to set $A$ by $\boldsymbol{\pi}$, and the supremum is over all possible initial distributions $\boldsymbol{q}$. The problem studied in this work is the construction of a non-trivial confidence interval $C_n = C_n(X_1, X_2, \dots, X_n, \delta) \subset [0, \infty]$, based only on the observed sample path $(X_1, X_2, \dots, X_n)$ and $\delta \in (0, 1)$, that succeeds with probability $1 - \delta$ in trapping the value of the mixing time $t_{\mathrm{mix}}$.

This problem is motivated by the numerous scientific applications and machine learning tasks in which the quantity of interest is the mean $\boldsymbol{\pi}(f) = \sum_i \pi_i f(i)$ for some function $f$ of the states of a Markov chain. This is the setting of the celebrated Markov Chain Monte Carlo (MCMC) paradigm [1], but the problem also arises in performance prediction involving time-correlated data, as is common in reinforcement learning [2]. Observable bounds on mixing times are useful in the

design and diagnostics of these methods; they yield effective approaches to assessing the estimation quality, even when *a priori* knowledge of the mixing time or correlation structure is unavailable.

**Main results.**   We develop the first procedure for constructing non-trivial and fully empirical confidence intervals for Markov mixing time. Consider a reversible ergodic Markov chain on $d$ states with absolute spectral gap $\gamma_\star$ and stationary distribution minorized by $\pi_\star$. As is well-known [3, Theorems 12.3 and 12.4],

$$(t_{\text{relax}} - 1)\ln 2 \ \leq \ t_{\text{mix}} \ \leq \ t_{\text{relax}}\ln\frac{4}{\pi_\star} \tag{2}$$

where $t_{\text{relax}} := 1/\gamma_\star$ is the *relaxation time*. Hence, it suffices to estimate $\gamma_\star$ and $\pi_\star$. Our main results are summarized as follows.

1. In Section 3.1, we show that in some problems $n = \Omega((d\log d)/\gamma_\star + 1/\pi_\star)$ observations are necessary for any procedure to guarantee constant multiplicative accuracy in estimating $\gamma_\star$ (Theorems 1 and 2). Essentially, in some problems *every* state may need to be visited about $\log(d)/\gamma_\star$ times, on average, before an accurate estimate of the mixing time can be provided, regardless of the actual estimation procedure used.

2. In Section 3.2, we give a point-estimator for $\gamma_\star$, and prove in Theorem 3 that it achieves multiplicative accuracy from *a single sample path* of length $\tilde{O}(1/(\pi_\star\gamma_\star^3))$.[1] We also provide a point-estimator for $\pi_\star$ that requires a sample path of length $\tilde{O}(1/(\pi_\star\gamma_\star))$. This establishes the feasibility of *estimating* the mixing time in this setting. However, the valid confidence intervals suggested by Theorem 3 depend on the unknown quantities $\pi_\star$ and $\gamma_\star$. We also discuss the importance of reversibility, and some possible extensions to non-reversible chains.

3. In Section 4, the construction of valid *fully empirical confidence intervals* for $\pi_\star$ and $\gamma_\star$ are considered. First, the difficulty of the task is explained, i.e., why the standard approach of turning the finite time confidence intervals of Theorem 3 into a fully empirical one fails. Combining several results from perturbation theory in a novel fashion we propose a new procedure and prove that it avoids slow convergence (Theorem 4). We also explain how to combine the empirical confidence intervals from Algorithm 1 with the non-empirical bounds from Theorem 3 to produce valid empirical confidence intervals. We prove in Theorem 5 that the width of these new intervals converge to zero asymptotically at least as fast as those from either Theorem 3 and Theorem 4.

**Related work.**   There is a vast statistical literature on estimation in Markov chains. For instance, it is known that under the assumptions on $(X_t)_t$ from above, the law of large numbers guarantees that the sample mean $\boldsymbol{\pi}_n(f) := \frac{1}{n}\sum_{t=1}^n f(X_t)$ converges almost surely to $\boldsymbol{\pi}(f)$ [4], while the central limit theorem tells us that as $n \to \infty$, the distribution of the deviation $\sqrt{n}(\boldsymbol{\pi}_n(f) - \boldsymbol{\pi}(f))$ will be normal with mean zero and asymptotic variance $\lim_{n\to\infty} n\operatorname{Var}(\boldsymbol{\pi}_n(f))$ [5].

Although these asymptotic results help us understand the limiting behavior of the sample mean over a Markov chain, they say little about the finite-time non-asymptotic behavior, which is often needed for the prudent evaluation of a method or even its algorithmic design [6–13]. To address this need, numerous works have developed Chernoff-type bounds on $\Pr(|\boldsymbol{\pi}_n(f) - \boldsymbol{\pi}(f)| > \epsilon)$, thus providing valuable tools for non-asymptotic probabilistic analysis [6, 14–16]. These probability bounds are larger than corresponding bounds for independent and identically distributed (iid) data due to the temporal dependence; intuitively, for the Markov chain to yield a fresh draw $X_{t'}$ that behaves as if it was independent of $X_t$, one must wait $\Theta(t_{\text{mix}})$ time steps. Note that the bounds generally depend on distribution-specific properties of the Markov chain (e.g., $\boldsymbol{P}$, $t_{\text{mix}}$, $\gamma_\star$), which are often unknown *a priori* in practice. Consequently, much effort has been put towards estimating these unknown quantities, especially in the context of MCMC diagnostics, in order to provide data-dependent assessments of estimation accuracy [e.g., 11, 12, 17–19]. However, these approaches generally only provide asymptotic guarantees, and hence fall short of our goal of empirical bounds that are valid with any finite-length sample path.

Learning with dependent data is another main motivation to our work. Many results from statistical learning and empirical process theory have been extended to sufficiently fast mixing, dependent

data [e.g., 20–26], providing learnability assurances (e.g., generalization error bounds). These results are often given in terms of mixing coefficients, which can be consistently estimated in some cases [27]. However, the convergence rates of the estimates from [27], which are needed to derive confidence bounds, are given in terms of unknown mixing coefficients. When the data comes from a Markov chain, these mixing coefficients can often be bounded in terms of mixing times, and hence our main results provide a way to make them fully empirical, at least in the limited setting we study.

It is possible to eliminate many of the difficulties presented above when allowed more flexible access to the Markov chain. For example, given a sampling oracle that generates independent transitions from any given state (akin to a "reset" device), the mixing time becomes an efficiently testable property in the sense studied in [28, 29]. On the other hand, when one only has a circuit-based description of the transition probabilities of a Markov chain over an exponentially-large state space, there are complexity-theoretic barriers for many MCMC diagnostic problems [30].

## 2 Preliminaries

### 2.1 Notations

We denote the set of positive integers by $\mathbb{N}$, and the set of the first $d$ positive integers $\{1, 2, \ldots, d\}$ by $[d]$. The non-negative part of a real number $x$ is $[x]_+ := \max\{0, x\}$, and $\lceil x \rceil_+ := \max\{0, \lceil x \rceil\}$. We use $\ln(\cdot)$ for natural logarithm, and $\log(\cdot)$ for logarithm with an arbitrary constant base. Bold-face symbols are used for vectors and matrices (e.g., $\boldsymbol{v}$, $\boldsymbol{M}$), and their entries are referenced by subindexing (e.g., $v_i$, $M_{i,j}$). For a vector $\boldsymbol{v}$, $\|\boldsymbol{v}\|$ denotes its Euclidean norm; for a matrix $\boldsymbol{M}$, $\|\boldsymbol{M}\|$ denotes its spectral norm. We use $\mathrm{Diag}(\boldsymbol{v})$ to denote the diagonal matrix whose $(i, i)$-th entry is $v_i$. The probability simplex is denoted by $\Delta^{d-1} = \{\boldsymbol{p} \in [0, 1]^d : \sum_{i=1}^d p_i = 1\}$, and we regard vectors in $\Delta^{d-1}$ as row vectors.

### 2.2 Setting

Let $\boldsymbol{P} \in (\Delta^{d-1})^d \subset [0, 1]^{d \times d}$ be a $d \times d$ row-stochastic matrix for an ergodic (i.e., irreducible and aperiodic) Markov chain. This implies there is a unique stationary distribution $\boldsymbol{\pi} \in \Delta^{d-1}$ with $\pi_i > 0$ for all $i \in [d]$ [3, Corollary 1.17]. We also assume that $\boldsymbol{P}$ is *reversible* (with respect to $\boldsymbol{\pi}$):

$$\pi_i P_{i,j} = \pi_j P_{j,i}, \quad i, j \in [d]. \tag{3}$$

The minimum stationary probability is denoted by $\pi_\star := \min_{i \in [d]} \pi_i$.

Define the matrices

$$\boldsymbol{M} := \mathrm{Diag}(\boldsymbol{\pi}) \boldsymbol{P} \quad \text{and} \quad \boldsymbol{L} := \mathrm{Diag}(\boldsymbol{\pi})^{-1/2} \boldsymbol{M} \, \mathrm{Diag}(\boldsymbol{\pi})^{-1/2}.$$

The $(i, j)$th entry of the matrix $M_{i,j}$ contains the *doublet probabilities* associated with $\boldsymbol{P}$: $M_{i,j} = \pi_i P_{i,j}$ is the probability of seeing state $i$ followed by state $j$ when the chain is started from its stationary distribution. The matrix $\boldsymbol{M}$ is symmetric on account of the reversibility of $\boldsymbol{P}$, and hence it follows that $\boldsymbol{L}$ is also symmetric. (We will strongly exploit the symmetry in our results.) Further, $\boldsymbol{L} = \mathrm{Diag}(\boldsymbol{\pi})^{1/2} \boldsymbol{P} \, \mathrm{Diag}(\boldsymbol{\pi})^{-1/2}$, hence $\boldsymbol{L}$ and $\boldsymbol{P}$ are similar and thus their eigenvalue systems are identical. Ergodicity and reversibility imply that the eigenvalues of $\boldsymbol{L}$ are contained in the interval $(-1, 1]$, and that 1 is an eigenvalue of $\boldsymbol{L}$ with multiplicity 1 [3, Lemmas 12.1 and 12.2]. Denote and order the eigenvalues of $\boldsymbol{L}$ as

$$1 = \lambda_1 > \lambda_2 \geq \cdots \geq \lambda_d > -1.$$

Let $\lambda_\star := \max\{\lambda_2, |\lambda_d|\}$, and define the (absolute) spectral gap to be $\gamma_\star := 1 - \lambda_\star$, which is strictly positive on account of ergodicity.

Let $(X_t)_{t \in \mathbb{N}}$ be a Markov chain whose transition probabilities are governed by $\boldsymbol{P}$. For each $t \in \mathbb{N}$, let $\boldsymbol{\pi}^{(t)} \in \Delta^{d-1}$ denote the marginal distribution of $X_t$, so

$$\boldsymbol{\pi}^{(t+1)} = \boldsymbol{\pi}^{(t)} \boldsymbol{P}, \quad t \in \mathbb{N}.$$

Note that the initial distribution $\boldsymbol{\pi}^{(1)}$ is arbitrary, and need not be the stationary distribution $\boldsymbol{\pi}$.

The goal is to estimate $\pi_\star$ and $\gamma_\star$ from the length $n$ sample path $(X_t)_{t \in [n]}$, and also to construct fully empirical confidence intervals that $\pi_\star$ and $\gamma_\star$ with high probability; in particular, the construction

of the intervals should not depend on any unobservable quantities, including $\pi_\star$ and $\gamma_\star$ themselves. As mentioned in the introduction, it is well-known that the *mixing time* of the Markov chain $t_{\mathrm{mix}}$ (defined in Eq. 1) is bounded in terms of $\pi_\star$ and $\gamma_\star$, as shown in Eq. (2). Moreover, convergence rates for empirical processes on Markov chain sequences are also often given in terms of mixing coefficients that can ultimately be bounded in terms of $\pi_\star$ and $\gamma_\star$ (as we will show in the proof of our first result). Therefore, valid confidence intervals for $\pi_\star$ and $\gamma_\star$ can be used to make these rates fully observable.

## 3 Point estimation

In this section, we present lower and upper bounds on achievable rates for estimating the spectral gap as a function of the length of the sample path $n$.

### 3.1 Lower bounds

The purpose of this section is to show lower bounds on the number of observations necessary to achieve a fixed multiplicative (or even just additive) accuracy in estimating the spectral gap $\gamma_\star$. By Eq. (2), the multiplicative accuracy lower bound for $\gamma_\star$ gives the same lower bound for estimating the mixing time. Our first result holds even for two state Markov chains and shows that a sequence length of $\Omega(1/\pi_\star)$ is necessary to achieve even a constant *additive* accuracy in estimating $\gamma_\star$.

**Theorem 1.** *Pick any $\bar{\pi} \in (0, 1/4)$. Consider any estimator $\hat{\gamma}_\star$ that takes as input a random sample path of length $n \leq 1/(4\bar{\pi})$ from a Markov chain starting from any desired initial state distribution. There exists a two-state ergodic and reversible Markov chain distribution with spectral gap $\gamma_\star \geq 1/2$ and minimum stationary probability $\pi_\star \geq \bar{\pi}$ such that*

$$\Pr\left[|\hat{\gamma}_\star - \gamma_\star| \geq 1/8\right] \geq 3/8.$$

Next, considering $d$ state chains, we show that a sequence of length $\Omega(d\log(d)/\gamma_\star)$ is required to estimate $\gamma_\star$ up to a constant multiplicative accuracy. Essentially, the sequence may have to visit all $d$ states at least $\log(d)/\gamma_\star$ times each, on average. This holds *even* if $\pi_\star$ is within a factor of two of the *largest* possible value of $1/d$ that it can take, i.e., when $\boldsymbol{\pi}$ is nearly uniform.

**Theorem 2.** *There is an absolute constant $c > 0$ such that the following holds. Pick any positive integer $d \geq 3$ and any $\bar{\gamma} \in (0, 1/2)$. Consider any estimator $\hat{\gamma}_\star$ that takes as input a random sample path of length $n < cd\log(d)/\bar{\gamma}$ from a $d$-state reversible Markov chain starting from any desired initial state distribution. There is an ergodic and reversible Markov chain distribution with spectral gap $\gamma_\star \in [\bar{\gamma}, 2\bar{\gamma}]$ and minimum stationary probability $\pi_\star \geq 1/(2d)$ such that*

$$\Pr\left[|\hat{\gamma}_\star - \gamma_\star| \geq \bar{\gamma}/2\right] \geq 1/4.$$

The proofs of Theorems 1 and 2 are given in Appendix A.[2]

### 3.2 A plug-in based point estimator and its accuracy

Let us now consider the problem of estimating $\gamma_\star$. For this, we construct a natural plug-in estimator. Along the way, we also provide an estimator for the minimum stationary probability, allowing one to use the bounds from Eq. (2) to trap the mixing time.

Define the random matrix $\widehat{\boldsymbol{M}} \in [0,1]^{d \times d}$ and random vector $\hat{\boldsymbol{\pi}} \in \Delta^{d-1}$ by

$$\widehat{M}_{i,j} := \frac{|\{t \in [n-1] : (X_t, X_{t+1}) = (i,j)\}|}{n-1}, \quad i,j \in [d],$$

$$\hat{\pi}_i := \frac{|\{t \in [n] : X_t = i\}|}{n}, \quad i \in [d].$$

Furthermore, define

$$\mathrm{Sym}(\widehat{\boldsymbol{L}}) := \frac{1}{2}(\widehat{\boldsymbol{L}} + \widehat{\boldsymbol{L}}^\top)$$

to be the symmetrized version of the (possibly non-symmetric) matrix

$$\widehat{\boldsymbol{L}} := \mathrm{Diag}(\hat{\boldsymbol{\pi}})^{-1/2} \widehat{\boldsymbol{M}} \, \mathrm{Diag}(\hat{\boldsymbol{\pi}})^{-1/2}.$$

Let $\hat{\lambda}_1 \geq \hat{\lambda}_2 \geq \cdots \geq \hat{\lambda}_d$ be the eigenvalues of $\mathrm{Sym}(\widehat{\boldsymbol{L}})$. Our estimator of the minimum stationary probability $\pi_\star$ is $\hat{\pi}_\star := \min_{i \in [d]} \hat{\pi}_i$, and our estimator of the spectral gap $\gamma_\star$ is $\hat{\gamma}_\star := 1 - \max\{\hat{\lambda}_2, |\hat{\lambda}_d|\}$.

These estimators have the following accuracy guarantees:

**Theorem 3.** *There exists an absolute constant $C > 0$ such that the following holds. Assume the estimators $\hat{\pi}_\star$ and $\hat{\gamma}_\star$ described above are formed from a sample path of length $n$ from an ergodic and reversible Markov chain. Let $\gamma_\star > 0$ denote the spectral gap and $\pi_\star > 0$ the minimum stationary probability. For any $\delta \in (0, 1)$, with probability at least $1 - \delta$,*

$$|\hat{\pi}_\star - \pi_\star| \leq C \left( \sqrt{\frac{\pi_\star \log \frac{d}{\pi_\star \delta}}{\gamma_\star n}} + \frac{\log \frac{d}{\pi_\star \delta}}{\gamma_\star n} \right) \tag{4}$$

*and*

$$|\hat{\gamma}_\star - \gamma_\star| \leq C \left( \sqrt{\frac{\log \frac{d}{\delta} \cdot \log \frac{n}{\pi_\star \delta}}{\pi_\star \gamma_\star n}} + \frac{\log \frac{1}{\gamma_\star}}{\gamma_\star n} \right) . \tag{5}$$

Theorem 3 implies that the sequence lengths required to estimate $\pi_\star$ and $\gamma_\star$ to within constant multiplicative factors are, respectively, $\tilde{O}\left(\frac{1}{\pi_\star \gamma_\star}\right)$ and $\tilde{O}\left(\frac{1}{\pi_\star \gamma_\star^3}\right)$. By Eq. (2), the second of these is also a bound on the required sequence length to estimate $t_{\mathrm{mix}}$.

The proof of Theorem 3 is based on analyzing the convergence of the sample averages $\widehat{\boldsymbol{M}}$ and $\hat{\boldsymbol{\pi}}$ to their expectation, and then using perturbation bounds for eigenvalues to derive a bound on the error of $\hat{\gamma}_\star$. However, since these averages are formed using a *single sample path* from a (possibly) non-stationary Markov chain, we cannot use standard large deviation bounds; moreover applying Chernoff-type bounds for Markov chains to each entry of $\widehat{\boldsymbol{M}}$ would result in a significantly worse sequence length requirement, roughly a factor of $d$ larger. Instead, we adapt probability tail bounds for sums of independent random matrices [32] to our non-iid setting by directly applying a blocking technique of [33] as described in the article of [20]. Due to ergodicity, the convergence rate can be bounded without any dependence on the initial state distribution $\boldsymbol{\pi}^{(1)}$. The proof of Theorem 3 is given in Appendix B.

Note that because the eigenvalues of $\boldsymbol{L}$ are the same as that of the transition probability matrix $\boldsymbol{P}$, we could have instead opted to estimate $\boldsymbol{P}$, say, using simple frequency estimates obtained from the sample path, and then computing the second largest eigenvalue of this empirical estimate $\widehat{\boldsymbol{P}}$. In fact, this approach is a way to extend to non-reversible chains, as we would no longer rely on the symmetry of $\boldsymbol{M}$ or $\boldsymbol{L}$. The difficulty with this approach is that $\boldsymbol{P}$ lacks the structure required by certain strong eigenvalue perturbation results. One could instead invoke the Ostrowski-Elsner theorem [cf. Theorem 1.4 on Page 170 of 34], which bounds the *matching distance* between the eigenvalues of a matrix $\boldsymbol{A}$ and its perturbation $\boldsymbol{A} + \boldsymbol{E}$ by $O(\|\boldsymbol{E}\|^{1/d})$. Since $\|\widehat{\boldsymbol{P}} - \boldsymbol{P}\|$ is expected to be of size $O(n^{-1/2})$, this approach will give a confidence interval for $\gamma_\star$ whose width shrinks at a rate of $O(n^{-1/(2d)})$—an exponential slow-down compared to the rate from Theorem 3. As demonstrated through an example from [34], the dependence on the $d$-th root of the norm of the perturbation cannot be avoided in general. Our approach based on estimating a symmetric matrix affords us the use of perturbation results that exploit more structure.

Returning to the question of obtaining a fully empirical confidence interval for $\gamma_\star$ and $\pi_\star$, we notice that, unfortunately, Theorem 3 falls short of being directly suitable for this, at least without further assumptions. This is because the deviation terms themselves depend inversely both on $\gamma_\star$ and $\pi_\star$, and hence can never rule out 0 (or an arbitrarily small positive value) as a possibility for $\gamma_\star$ or $\pi_\star$.[3] In effect, the fact that the Markov chain could be slow mixing and the long-term frequency of some

**Algorithm 1** Empirical confidence intervals

---

**Input**: Sample path $(X_1, X_2, \dots, X_n)$, confidence parameter $\delta \in (0, 1)$.

1: Compute state visit counts and smoothed transition probability estimates:
$$N_i := |\{t \in [n-1] : X_t = i\}|, \quad i \in [d];$$
$$N_{i,j} := |\{t \in [n-1] : (X_t, X_{t+1}) = (i,j)\}|, \quad \widehat{P}_{i,j} := \frac{N_{i,j} + 1/d}{N_i + 1}, \quad (i,j) \in [d]^2.$$

2: Let $\widehat{A}^\#$ be the group inverse of $\widehat{A} := I - \widehat{P}$.

3: Let $\hat{\boldsymbol{\pi}} \in \Delta^{d-1}$ be the unique stationary distribution for $\widehat{P}$.

4: Compute eigenvalues $\hat{\lambda}_1 \geq \hat{\lambda}_2 \geq \cdots \geq \hat{\lambda}_d$ of $\mathrm{Sym}(\widehat{L})$, where $\widehat{L} := \mathrm{Diag}(\hat{\boldsymbol{\pi}})^{1/2} \widehat{P} \, \mathrm{Diag}(\hat{\boldsymbol{\pi}})^{-1/2}$.

5: Spectral gap estimate:
$$\hat{\gamma}_\star := 1 - \max\{\hat{\lambda}_2, |\hat{\lambda}_d|\}.$$

6: Empirical bounds for $|\widehat{P}_{i,j} - P_{i,j}|$ for $(i,j) \in [d]^2$: $c := 1.01$, $\tau_{n,\delta} := \inf\{t \geq 0 : 2d^2(1 + \lceil \log_c \frac{2n}{t} \rceil_+)e^{-t} \leq \delta\}$,

and $\quad \widehat{B}_{i,j} := \left( \sqrt{\frac{c\tau_{n,\delta}}{2N_i}} + \sqrt{\frac{c\tau_{n,\delta}}{2N_i} + \sqrt{\frac{2c\widehat{P}_{i,j}(1 - \widehat{P}_{i,j})\tau_{n,\delta}}{N_i}} + \frac{(5/3)\tau_{n,\delta} + |\widehat{P}_{i,j} - 1/d|}{N_i}} \right)^2.$

7: Relative sensitivity of $\boldsymbol{\pi}$:
$$\hat{\kappa} := \frac{1}{2} \max \left\{ \widehat{A}^\#_{j,j} - \min \left\{ \widehat{A}^\#_{i,j} : i \in [d] \right\} : j \in [d] \right\}.$$

8: Empirical bounds for $\max_{i \in [d]} |\hat{\pi}_i - \pi_i|$ and $\max \bigcup_{i \in [d]} \{|\sqrt{\pi_i/\hat{\pi}_i} - 1|, |\sqrt{\hat{\pi}_i/\pi_i} - 1|\}$:
$$\hat{b} := \hat{\kappa} \max \left\{ \widehat{B}_{i,j} : (i,j) \in [d]^2 \right\}, \qquad \hat{\rho} := \frac{1}{2} \max \bigcup_{i \in [d]} \left\{ \frac{\hat{b}}{\hat{\pi}_i}, \frac{\hat{b}}{[\hat{\pi}_i - \hat{b}]_+} \right\}.$$

9: Empirical bounds for $|\hat{\gamma}_\star - \gamma_\star|$:
$$\hat{w} := 2\hat{\rho} + \hat{\rho}^2 + (1 + 2\hat{\rho} + \hat{\rho}^2) \left( \sum_{(i,j) \in [d]^2} \frac{\hat{\pi}_i}{\hat{\pi}_j} \hat{B}_{i,j}^2 \right)^{1/2}.$$

---

states could be small makes it difficult to be confident in the estimates provided by $\hat{\gamma}_\star$ and $\hat{\pi}_\star$. This suggests that in order to obtain fully empirical confidence intervals, we need an estimator that is not subject to such effects—we pursue this in Section 4. Theorem 3 thus primarily serves as a point of comparison for what is achievable in terms of estimation accuracy when one does not need to provide empirical confidence bounds.

## 4 Fully empirical confidence intervals

In this section, we address the shortcoming of Theorem 3 and give fully empirical confidence intervals for the stationary probabilities and the spectral gap $\gamma_\star$. The main idea is to use the Markov property to eliminate the dependence of the confidence intervals on the unknown quantities (including $\pi_\star$ and $\gamma_\star$). Specifically, we estimate the transition probabilities from the sample path using simple frequency estimates: as a consequence of the Markov property, for each state, the frequency estimates converge at a rate that depends only on the number of visits to the state, and in particular the rate (given the visit count of the state) is independent of the mixing time of the chain.

As discussed in Section 3, it is possible to form a confidence interval for $\gamma_\star$ based on the eigenvalues of an estimated transition probability matrix by appealing to the Ostrowski-Elsner theorem. However, as explained earlier, this would lead to a slow $O(n^{-1/(2d)})$ rate. We avoid this slow rate by using an estimate of the symmetric matrix $\boldsymbol{L}$, so that we can use a stronger perturbation result (namely Weyl's inequality, as in the proof of Theorem 3) available for symmetric matrices.

To form an estimate of $\boldsymbol{L}$ based on an estimate of the transition probabilities, one possibility is to estimate $\boldsymbol{\pi}$ using a frequency-based estimate for $\boldsymbol{\pi}$ as was done in Section 3, and appeal to the relation $\boldsymbol{L} = \mathrm{Diag}(\boldsymbol{\pi})^{1/2} \boldsymbol{P} \mathrm{Diag}(\boldsymbol{\pi})^{-1/2}$ to form a plug-in estimate. However, as noted in Section 3.2, confidence intervals for the entries of $\boldsymbol{\pi}$ formed this way may depend on the mixing time. Indeed, such an estimate of $\boldsymbol{\pi}$ does not exploit the Markov property.

We adopt a different strategy for estimating $\boldsymbol{\pi}$, which leads to our construction of empirical confidence intervals, detailed in Algorithm 1. We form the matrix $\widehat{\boldsymbol{P}}$ using smoothed frequency estimates of $\boldsymbol{P}$ (Step 1), then compute the so-called group inverse $\widehat{\boldsymbol{A}}^{\#}$ of $\widehat{\boldsymbol{A}} = \boldsymbol{I} - \widehat{\boldsymbol{P}}$ (Step 2), followed by finding the unique stationary distribution $\hat{\boldsymbol{\pi}}$ of $\widehat{\boldsymbol{P}}$ (Step 3), this way decoupling the bound on the accuracy of $\hat{\boldsymbol{\pi}}$ from the mixing time. The group inverse $\widehat{\boldsymbol{A}}^{\#}$ of $\widehat{\boldsymbol{A}}$ is uniquely defined; and if $\widehat{\boldsymbol{P}}$ defines an ergodic chain (which is the case here due to the use of the smoothed estimates), $\widehat{\boldsymbol{A}}^{\#}$ can be computed at the cost of inverting an $(d-1)\times(d-1)$ matrix [35, Theorem 5.2].[4] Further, once given $\widehat{\boldsymbol{A}}^{\#}$, the unique stationary distribution $\hat{\boldsymbol{\pi}}$ of $\widehat{\boldsymbol{P}}$ can be read out from the last row of $\widehat{\boldsymbol{A}}^{\#}$ [35, Theorem 5.3]. The group inverse is also be used to compute the sensitivity of $\boldsymbol{\pi}$. Based on $\hat{\boldsymbol{\pi}}$ and $\widehat{\boldsymbol{P}}$, we construct the plug-in estimate $\widehat{\boldsymbol{L}}$ of $\boldsymbol{L}$, and use the eigenvalues of its symmetrization to form the estimate $\hat{\gamma}_\star$ of the spectral gap (Steps 4 and 5). In the remaining steps, we use perturbation analyses to relate $\hat{\boldsymbol{\pi}}$ and $\boldsymbol{\pi}$, viewing $\boldsymbol{P}$ as the perturbation of $\widehat{\boldsymbol{P}}$; and also to relate $\hat{\gamma}_\star$ and $\gamma_\star$, viewing $\boldsymbol{L}$ as a perturbation of $\mathrm{Sym}(\widehat{\boldsymbol{L}})$. Both analyses give error bounds entirely in terms of observable quantities (e.g., $\hat{\kappa}$), tracing back to empirical error bounds for the smoothed frequency estimates of $\boldsymbol{P}$.

The most computationally expensive step in Algorithm 1 is the computation of the group inverse $\widehat{\boldsymbol{A}}^{\#}$, which, as noted reduces to matrix inversion. Thus, with a standard implementation of matrix inversion, the algorithm's time complexity is $O(n + d^3)$, while its space complexity is $O(d^2)$.

To state our main theorem concerning Algorithm 1, we first define $\kappa$ to be analogous to $\hat{\kappa}$ from Step 7, with $\widehat{\boldsymbol{A}}^{\#}$ replaced by the group inverse $\boldsymbol{A}^{\#}$ of $\boldsymbol{A} := \boldsymbol{I} - \boldsymbol{P}$. The result is as follows.

**Theorem 4.** *Suppose Algorithm 1 is given as input a sample path of length $n$ from an ergodic and reversible Markov chain and confidence parameter $\delta \in (0,1)$. Let $\gamma_\star > 0$ denote the spectral gap, $\boldsymbol{\pi}$ the unique stationary distribution, and $\pi_\star > 0$ the minimum stationary probability. Then, on an event of probability at least $1 - \delta$,*

$$\pi_i \in [\hat{\pi}_i - \hat{b}, \hat{\pi}_i + \hat{b}] \quad \text{for all } i \in [d], \qquad \text{and} \qquad \gamma_\star \in [\hat{\gamma}_\star - \hat{w}, \hat{\gamma}_\star + \hat{w}].$$

*Moreover, $\hat{b}$ and $\hat{w}$ almost surely satisfy (as $n \to \infty$)*

$$\hat{b} = O\left( \max_{(i,j)\in[d]^2} \kappa \sqrt{\frac{P_{i,j} \log\log n}{\pi_i n}} \right), \quad \hat{w} = O\left( \frac{\kappa}{\pi_\star} \sqrt{\frac{\log\log n}{\pi_\star n}} + \sqrt{\frac{d \log\log n}{\pi_\star n}} \right).\text{[5]}$$

The proof of Theorem 4 is given in Appendix C. As mentioned above, the obstacle encountered in Theorem 3 is avoided by exploiting the Markov property. We establish fully observable upper and lower bounds on the entries of $\boldsymbol{P}$ that converge at a $\sqrt{n/\log\log n}$ rate using standard martingale tail inequalities; this justifies the validity of the bounds from Step 6. Properties of the group inverse [35, 36] and eigenvalue perturbation theory [34] are used to validate the empirical bounds on $\pi_i$ and $\gamma_\star$ developed in the remaining steps of the algorithm.

The first part of Theorem 4 provides valid empirical confidence intervals for each $\pi_i$ and for $\gamma_\star$, which are simultaneously valid at confidence level $\delta$. The second part of Theorem 4 shows that the

width of the intervals decrease as the sequence length increases. We show in Appendix C.5 that $\kappa \leq d/\gamma_\star$, and hence $\hat{b} = O\left(\max_{(i,j)\in[d]^2} \frac{d}{\gamma_\star}\sqrt{\frac{P_{i,j}\log\log n}{\pi_i n}}\right)$, $\hat{w} = O\left(\frac{d}{\pi_\star\gamma_\star}\sqrt{\frac{\log\log n}{\pi_\star n}}\right)$.

It is easy to combine Theorems 3 and 4 to yield intervals whose widths shrink at least as fast as both the non-empirical intervals from Theorem 3 and the empirical intervals from Theorem 4. Specifically, determine lower bounds on $\pi_\star$ and $\gamma_\star$ using Algorithm 1, $\pi_\star \geq \min_{i\in[d]}[\hat{\pi}_i - \hat{b}]_+$, $\gamma_\star \geq [\hat{\gamma}_\star - \hat{w}]_+$; then plug-in these lower bounds for $\pi_\star$ and $\gamma_\star$ in the deviation bounds in Eq. (5) from Theorem 3. This yields a new interval centered around the estimate of $\gamma_\star$ from Theorem 3, and it no longer depends on unknown quantities. The interval is a valid $1 - 2\delta$ probability confidence interval for $\gamma_\star$, and for sufficiently large $n$, the width shrinks at the rate given in Eq. (5). We can similarly construct an empirical confidence interval for $\pi_\star$ using Eq. (4), which is valid on the same $1 - 2\delta$ probability event.[6] Finally, we can take the intersection of these new intervals with the corresponding intervals from Algorithm 1. This is summarized in the following theorem, which we prove in Appendix D.

**Theorem 5.** *The following holds under the same conditions as Theorem 4. For any $\delta \in (0,1)$, the confidence intervals $\widehat{U}$ and $\widehat{V}$ described above for $\pi_\star$ and $\gamma_\star$, respectively, satisfy $\pi_\star \in \widehat{U}$ and $\gamma_\star \in \widehat{V}$ with probability at least $1 - 2\delta$. Furthermore, the widths of these intervals almost surely satisfy (as $n \to \infty$) $|\widehat{U}| = O\left(\sqrt{\frac{\pi_\star \log\frac{d}{\pi_\star\delta}}{\gamma_\star n}}\right)$, $|\widehat{V}| = O\left(\min\left\{\sqrt{\frac{\log\frac{d}{\delta}\cdot\log(n)}{\pi_\star\gamma_\star n}}, \hat{w}\right\}\right)$, where $\hat{w}$ is the width from Algorithm 1.*

## 5  Discussion

The construction used in Theorem 5 applies more generally: Given a confidence interval of the form $I_n = I_n(\gamma_\star, \pi_\star, \delta)$ for some confidence level $\delta$ and a fully empirical confidence set $E_n(\delta)$ for $(\gamma_\star, \pi_\star)$ for the same level, $I'_n = E_n(\delta) \cap \cup_{(\gamma,\pi)\in E_n(\delta)} I_n(\gamma, \pi, \delta)$ is a valid fully empirical $2\delta$-level confidence interval whose asymptotic width matches that of $I_n$ up to lower order terms under reasonable assumptions on $E_n$ and $I_n$. In particular, this suggests that future work should focus on closing the gap between the lower and upper bounds on the accuracy of point-estimation. Another interesting direction is to reduce the computation cost: The current cubic cost in the number of states can be too high even when the number of states is only moderately large.

Perhaps more important, however, is to extend our results to large state space Markov chains: In most practical applications the state space is continuous or is exponentially large in some natural parameters. As follows from our lower bounds, without further assumptions, the problem of fully data dependent estimation of the mixing time is intractable for information theoretical reasons. Interesting directions for future work thus must consider Markov chains with specific structure. Parametric classes of Markov chains, including but not limited to Markov chains with factored transition kernels with a few factors, are a promising candidate for such future investigations. The results presented here are a first step in the ambitious research agenda outlined above, and we hope that they will serve as a point of departure for further insights in the area of fully empirical estimation of Markov chain parameters based on a single sample path.

## Footnotes

[1] The $\tilde{O}(\cdot)$ notation suppresses logarithmic factors.

[2]A full version of this paper, with appendices, is available on arXiv [31].

[3]Using Theorem 3, it is possible to trap $\gamma_\star$ in the union of *two* empirical confidence intervals—one around $\hat{\gamma}_\star$ and the other around zero, both of which shrink in width as the sequence length increases.

[4] The group inverse of a square matrix $\boldsymbol{A}$, a special case of the *Drazin inverse*, is the unique matrix $\boldsymbol{A}^{\#}$ satisfying $\boldsymbol{A}\boldsymbol{A}^{\#}\boldsymbol{A} = \boldsymbol{A}$, $\boldsymbol{A}^{\#}\boldsymbol{A}\boldsymbol{A}^{\#} = \boldsymbol{A}^{\#}$ and $\boldsymbol{A}^{\#}\boldsymbol{A} = \boldsymbol{A}\boldsymbol{A}^{\#}$.

[5] In Theorems 4 and 5, our use of big-$O$ notation is as follows. For a random sequence $(Y_n)_n$ and a (non-random) positive sequence $(\varepsilon_{\theta,n})_n$ parameterized by $\theta$, we say "$Y_n = O(\varepsilon_{\theta,n})$ holds almost surely as $n \to \infty$" if there is some universal constant $C > 0$ such that for all $\theta$, $\limsup_{n\to\infty} Y_n/\varepsilon_{\theta,n} \leq C$ holds almost surely.

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
