[Reviews · NeurIPS 2015]

Submitted by Assigned_Reviewer_1

1. The paper would be much more interesting to the NIPS audience if the authors provided an application of their work to a practical machine learning example, e.g., MCMC for reinforcement learning (as stated for motivation in the Introduction) or for marginalization of a graphical model. Without such an example, which would establish tightness of the proposed bounds and confidence intervals, it is difficult to appreciate the practical utility of the theoretical results presented in the paper.

2. After Theorem 1, in the statement that each state may have to be visited \log(d)/\gamma_* times on average, "at least" should be added considering the non-uniformity of the state frequencies.

3. In Theorems 1 and 2, since a lower bound on the number of observations is sought, it is more appropriate to present the results for n \geq c/\overline{\pi} and n \geq cd log(d)/\overline{\gamma}, instead of the reversed inequalities in these theorem statements. If the authors adopted this they would obtain upper bounds instead of lower bounds, on the error probabilities shown in each theorem.

4. It is not clear how (4) and (5) in Theorem 3 imply n \tilde{O}\left(\frac{1}{\pi_* \gamma_*} \right) and

n \tilde{O}\left(\frac{1}{\pi_* \gamma_*^3} \right), respectively. Indeed, one can easily see (suppressing the logarithmic terms) that (4) implies

n \tilde{O}\left(\max\left\{\frac{\pi_*}{\gamma_*}, \frac{1}{\gamma_*} \right\} \right) \tilde{O}\left(\frac{1}{\gamma_*} \right)

and similarly, that (5) implies

n \tilde{O}\left(\max\left\{\frac{\pi_*\gamma_*}{\gamma_*}, \frac{1}{\gamma_*} \right\} \right) \tilde{O}\left(\frac{1}{\pi_*\gamma_*} \right)

However, these do not imply the desired result. More explanation is needed.

5. The lower bound on the number of samples stated in Main Results is not the same as the lower bound in Theorem 2. The former lower bound is \Omega((d\log d)/\gamma_*+1/\pi_*) while the latter lower bound is missing the term 1/\pi_*. Explanation of this discrepancy is required.

6. There seems to be a problem with the proof of Theorem 1 in Appendix B. The following counterexample violates the claims in the proof. Let \overline{\pi} 1/4 for which \epsilon 2/3 and \lambda_* -1/6 < 0, which violates the definition \lambda_* \max(\lambda_2,|\lambda_d|). Furthermore, for n < c(2+\epsilon)/\epsilon 4c, choosing c 1, for which n 3, we compute the probability of staying in state 1 as (1-\epsilon)^n 1/27 which is less than the presented result of 1/4.

7. The last sentence of the proof of Thm 2 requires elaboration. It is not clear how it follows.

Minor comments

1. In the first paragraph of the Related Work the authors need clearly state the assumptions on the Markov chain, e.g. reversible, ergodic, aperiodic, irreducible, finite-state.

2. The title of Section 3 is misleading since they are not performing "point estimation" but rather interval estimation.

3. The authors repeated claims of obtaining confidence intervals that are "empirical" and "based on a single sample path" are a overemphasized. Any useful confidence interval must be empirical and by definition, a confidence interval is an interval estimate, and always depends on a given sample.

Summary: The paper proposes a non-asymptotic confidence interval for the mixing parameter of stationary Markov chain. The proofs in the paper lack clarity and the paper suffers from an absence of practical machine learning examples of the theory.

Submitted by Assigned_Reviewer_2

The paper considers the problem of estimating the mixing time of a reversible ergodic Markov chain from a sample path. This is tied to estimation of the spectral gap \gamma* and minimum stationary probability \pi*. The main results are: 1 lower bounds on the sample complexity of estimation of \gamma* and \pi* in terms of lower bounds on these quantities. 2 bounds on deviations of natural (plug-in) estimators of \gamma* and \pi* in terms of these quantities. 3 an algorithm for estimates and confidence intervals for \gamma* and \pi*, based on smoothed estimates and confidence intervals for transition probabilities. 4 an approach to computing confidence intervals for the plug-in estimates, based on lower bounds on \gamma* and \pi* that follow from the more refined estimates.

These are worthwhile contributions to a problem that is very significant. The techniques are novel: they involve working with a matrix that is similar to the transition probability matrix but is symmetric in the reversible case because of detailed balance.

This simplifies analysis of perturbations from estimation error.

The paper was mostly well-written, although adding more detail would have improved the appendix. The motivation for Theorem 5 (computing confidence intervals for the simple esimator using lower bounds from the more complex estimator) is unclear. When will it lead to a better estimate than the complex estimator? What is the advantage?

Minor points:

line 155: "contain" is missing.

line 188: Theorem 2 is false as stated. Consider the estimator \hat\gamma=3\bar\gamma/2. The error is in the claim on line 956, which is missing a constant factor.

line 232: It is standard to use \tilde O to suppress log factors of quantities that are already included in polynomials in the expression. It seems misleading to suppress quantities that are not included at all (e.g., log d factors). (Everything can be written as the log of a reparameterized quantity.)

line 318: kappa is not defined until the appendix (line 1090). (Or else I missed the definition?)

line 521: would help to give a more precise reference to this inequality than just a book.

line 538: adding two inequalities to this display would make it much clearer without taking up any space.

line 549: Include more detail - this equality is tedious to check. Why push that burden onto the reader?

line 562: why is the inequality true?

line 926: there's a mistake in pi^(2): the probabilities are reversed.

line 958: this claim is not clear. Working with this restricted family of Markov chains, it's only necessary to find an i,j pair such that P_{i,j}=\epsilon'/(d-1). Of course, a similar argument will work.
Summary: New sample-based estimates and confidence intervals for the mixing time of a reversible ergodic Markov chain. Interesting results and techniques, well-written paper. Worthwhile contribution.

Submitted by Assigned_Reviewer_3

The paper investigates the interesting question of estimating the mixing time by observing a single sample path of the irreducible, reversible Markov chain with finite state space. The paper is well written and easy to read. The first sentence of the abstract seems to over-claim the contributions of the paper: The mixing time is not estimated at a prescribed confidence level; the authors provide upper and lower bounds on the spectral gap and the lower bound of the stationary distribution, which are well known "proxies" of the mixing time. These bounds are then plugged in (2), which inevitably induces inaccuracies in the mixing time estimate.

The authors first present in Section 3 a simple way of estimating the spectral gap and the minimum stationary distribution. They derive corresponding confidence intervals, which depends on the (unknown) Markov chain parameters. In Section 4, the authors provide new estimators and manage to derive fully empirical confidence intervals, which is very appealing in practice. The empirical confidence intervals are given in the pseudo-code on Algorithm 1, which is a bit odd. When reading Theorem 4, \hat{b} and \hat{w} have not been introduced, and in view of the conditions on these two numbers given in the theorem, we first have the impression that the confidence bounds still depend on the Markov chain parameters ... This is confusing. The authors could for example state that \hat{b} and \hat{w} are derived from the data only, and given in Algorithm 1.

Overall, the paper is enjoyable, but I would have liked to see at least a few numerical experiments to assess the practical performance of the setimators, and importantly on the empirical confidence intervals (are they tight, very loose?).
Summary: The paper presents methods to estimate the spectral gap and the minimum state probability of a finite state, irreducible, and reversible Markov chain. These two estimated quantities can in turn be used to further estimate the mixing time of the chain. The paper provides interesting results, but their presentation could be improved.

Author Feedback
Author rebuttal: We thank the referees for the helpful comments. We'll incorporate the suggestions in a revision of the paper, especially where more detailed explanations are needed.

The following two concerns were common to multiple reviews.

Tightness of bounds: We don't claim tightness of the bounds (we did not attempt to optimize the constants) but our results give the first non-trivial (upper and lower) bounds in this challenging problem. In particular, the estimator is based on a less than straighforward, indirect approach while avoiding slow rates (as explained at the beginning of Section 4) that builds on advanced perturbation results from the theory of stochastic matrices. Our work is motivated by practical applications where fully data-dependent intervals were sought but not actually achieved. Our results are a first step towards a practical solution: we demonstrate feasibility and also prove limitations on what is possible, both add significantly to our knowledge.

Relaxation vs mixing time: In the revision, we will be more explicit about estimating the relaxation time rather than mixing time proper (throughout the abstract and paper). We fully agree that this is a very important distinction that should be emphasized, and we'll amend the overstated claims throughout.

Additional responses to individual reviews follow.

Reviewer_1

2. We'll make this edit in the revision.

3. Qualitatively, Theorems 1 and 2 show that if n is less than some function of (d,pi,gamma), then for any estimator based on a sample path of length n, there is some Markov chain such that the error is large. We'll add additional clarifying interpretations in the revision.

4. If n > max{16C^2,4C}/(pi_* gamma_*), then the RHS in (4) (without the logarithmic factor) is less than pi_*/2.

If n > max{16C^2,4C}/(pi_* gamma_*^3), then the RHS in (5) (without the logarithmic factor) is less than gamma_*/2.

We will add these details in the revision.

5. Combining Theorems 1 and 2 with a standard argument yields the lower bound in the introduction. We will include the details in the revision.

6. Regarding Theorem 1: Note that "c" is an absolute constant in the theorem, not something that can be chosen arbitrarily. In particular, one value that works in the theorem is c=1/4.

There were some minor numerical calculation errors in the proof of Theorem 1, which we can clarify here. We need to change the "1/6" in the theorem statement and proof to "1/8". Furthermore, in the proof, \bar\pi should be taken in (0,1/4), and we should set \varepsilon := \bar\pi. Observe that \pi^{(2)} = (1/(1+2\varepsilon),2\varepsilon/(1+2\varepsilon)). Thus, \pi_\star \geq \bar\pi in both P^{(1)} and P^{(2)}. To guarantee |\hat\gamma_\star - \gamma_\star| < 1/8, it must be possible to distinguish the two Markov chains. The rest of the proof is correct as is.

7. The conclusion of Theorem 2 follows using the second-moment method (e.g., Paley-Zygmund) and the moment bounds that, with probability > 1/4, the earliest time in which a sample path visits all d states is at least c d log(d) / \bar\gamma. This implies that an estimator using a shorter sample path cannot distinguish P^0 from P^i for some i (with probability > 1/4), and hence must have large estimation error. We'll include these details in the revision.

Minor comment 3: We respectfully disagree. The definition of a confidence interval does not distinguish intervals based on what is known, which itself may depend on the application (in specific applications additional knowledge if often available).

Reviewer_2

Theorem 5: We get an improvement when the min in the bound for |\hat{V}| is achieved by the first term. A sheepish caveat: this is not always the case for all sufficiently large n, due to the sqrt{log(n)} term in the numerator, but will be true for a very large range of n. There is more discussion in Section 5.

Theorem 2 (line 188): Indeed, including an extra constant factor in the claim resolves this issue.

\tilde{O}: We'll adopt this more standard use in the revision.

kappa: It is defined just above Theorem 4.

line 926: Thanks. See response to Issue 6 by Reviewer 1.

Theorem 2 (line 958): An estimator cannot rule out P^i without having visited state i. We'll clarify this point in the revision.

Reviewer_7

Perfect sampling: Indeed, this line of work is related, although these methods generally do not apply in our setting of estimation from a single sample path. We'll discuss this in the revision.